# Comparing Local and Systemic Control between Partial- and Whole-Breast Radiotherapy in Low-Risk Breast Cancer—A Meta-Analysis of Randomized Trials

**DOI:** 10.3390/cancers13122967

**Published:** 2021-06-13

**Authors:** Jan Haussmann, Wilfried Budach, Vratislav Strnad, Stefanie Corradini, David Krug, Livia Schmidt, Balint Tamaskovics, Edwin Bölke, Ioannis Simiantonakis, Kai Kammers, Christiane Matuschek

**Affiliations:** 1Department of Radiation Oncology, Heinrich Heine University, 40225 Dusseldorf, Germany; Jan.Haussmann@med.uni-duesseldorf.de (J.H.); Wilfried.Budach@med.uni-duesseldorf.de (W.B.); Livia.Schmidt@med.uni-duesseldorf.de (L.S.); Balint.tamaskovics@med.uni-duesseldorf.de (B.T.); Ioannis.Simiantonakis@med.uni-duesseldorf.de (I.S.); matuschek@med.uni-duesseldorf.de (C.M.); 2Department of Radiation Oncology, University Erlangen, 91054 Erlangen, Germany; vratislav.strnad@uk-erlangen.de; 3Department of Radiation Oncology, University Hospital LMU (Ludwig Maximillian), 81377 Munich, Germany; Stefanie.Corradini@med.uni-muenchen.de; 4Department of Radiation Oncology, University Hospital Schleswig-Holstein, 24105 Kiel, Germany; david.krug@uksh.de; 5Division of Biostatistics and Bioinformatics, Department of Oncology, The Sidney Kimmel Comprehensive Cancer Center at Johns Hopkins, The Johns Hopkins University School of Medicine, Baltimore, MD 21205, USA; kai.kammers@jhu.edu

**Keywords:** breast cancer, radiotherapy, partial-breast treatment, local recurrence

## Abstract

**Simple Summary:**

This meta-analysis compares the treatment results of partial-breast radiotherapy to those of whole-breast radiotherapy after breast conserving surgery in early-stage breast cancer. The results show that the tumor is slightly more likely to recur in the operated breast after partial radiotherapy compared to radiation therapy to the whole breast. These additional recurrences are located away from the original tumor bed. The technique by which partial-breast radiotherapy is applied also appears to affect the likeliness of tumor regrowth. Intraoperative radiation, given during the removal of the tumor, might lead to more relapses compared to other techniques. Partial-breast treatment also led to more lymph node recurrences in a very small number of patients. However, rates of distant relapses were not increased. We were unable to identify a specific subgroup that was most suitable for partial-breast irradiation. The differences between treatment of partial- and whole-breast radiotherapy are small when the patient groups and the radiation technique are appropriately selected.

**Abstract:**

Purpose/Objective: The standard treatment for localized low-risk breast cancer is breast-conserving surgery, followed by adjuvant radiotherapy and appropriate systemic therapy. As the majority of local recurrences occur at the site of the primary tumor, numerous trials have investigated partial-breast irradiation (PBI) instead of whole-breast treatment (WBI) using a multitude of irradiation techniques and fractionation regimens. This meta-analysis addresses the impact on disease-specific endpoints, such as local and regional control, as well as disease-free survival of PBI compared to that of WBI in published randomized trials. Material and Methods: We conducted a systematic literature review and searched for randomized trials comparing WBI and PBI in early-stage breast cancer with publication dates after 2009. The meta-analysis was based on the published event rates and the effect sizes for available oncological endpoints of at least two trials reporting on them. We evaluated in-breast tumor recurrences (IBTR), local recurrences at the primary site and elsewhere in the ipsilateral breast, regional recurrences (RR), distant metastasis-free interval (DMFI), disease-free survival (DFS), contralateral breast cancer (CBC), and second primary cancer (SPC). Furthermore, we aimed to assess the impact of different PBI techniques and subgroups on IBTR. We performed all statistical analyses using the inverse variance heterogeneity model to pool effect sizes. Results: For the intended meta-analysis, we identified 13 trials (overall 15,561 patients) randomizing between PBI and WBI. IBTR was significantly higher after PBI (OR = 1.66; CI-95%: 1.07–2.58; *p* = 0.024) with an absolute difference of 1.35%. We detected significant heterogeneity in the analysis of the PBI technique with intraoperative radiotherapy resulting in higher local relapse rates (OR = 3.67; CI-95%: 2.28–5.90; *p* < 0.001). Other PBI techniques did not show differences to WBI in IBTR. Both strategies were equally effective at the primary tumor site, but PBI resulted in statistically more IBTRs elsewhere in the ipsilateral breast. IBTRs after WBI were more likely to be located at the primary tumor bed, whereas they appeared equally distributed within the breast after PBI. RR was also more frequent after PBI (OR = 1.75; CI-95%: 1.07–2.88; *p* < 0.001), yet we did not detect any differences in DMFI (OR = 1.08; CI-95%: 0.89–1.30; *p* = 0.475). DFS was significantly longer in patients treated with WBI (OR = 1.14; CI-95%: 1.02–1.27; *p* = 0.003). CBC and SPC were not different in the test groups (OR = 0.81; CI-95%: 0.65–1.01; *p* = 0.067 and OR = 1.09; CI-95%: 0.85–1.40; *p* = 0.481, respectively). Conclusion: Limiting the target volume to partial-breast radiotherapy appears to be appropriate when selecting patients with a low risk for local and regional recurrences and using a suitable technique.

## 1. Introduction

Whole-breast irradiation (WBI) and adequate systemic therapy are the two standard treatments after breast conserving surgery of early-stage breast cancer. This multimodal approach has been shown to be the oncological equivalent to mastectomy in numerous randomized trials [1,2,3,4,5]. Both adjuvant treatment modalities have been shown to reduce recurrence rates and improve overall survival [6,7]. With the advent of more sophisticated radiological modalities, standardized pathological testing, low-morbidity surgery, and effective systemic therapies, attempts have been made to de-escalate the treatment in early-stage breast cancer. Omission of adjuvant whole-breast irradiation was studied in multiple randomized trials of low-risk breast cancer patients [8,9,10,11,12,13,14,15]. Two meta-analyses demonstrated that omission of WBI had no negative impact on overall survival in selected patients but led to a significant loss in local control [16,17].

Numerous reports showed that the majority of local recurrences occur at the primary tumor bed after WBI [2,18,19,20,21,22,23,24,25,26,27,28]. Furthermore, histopathological analyses of mastectomy specimen showed that the highest density of tumor tissue was found within the first 2 cm around the tumor [29]. These observations led to the introduction of the de-escalation approach of partial-breast irradiation (PBI). Here, the treated breast volume is restricted to the tumor bed and the directly surrounding tissue.

The concept of PBI aimed to achieve non-inferior or equivalent control rates, improve cosmetic results, and reduce toxicities. As PBI treats a smaller volume of breast tissue, it has also been assumed that an acceleration of the treatment schedule might be possible to optimize convenience for the patients. However, initial randomized trials showed higher recurrence rates, and the differences between the treatments seemed to be highly dependent on the included risk groups [30,31].

We attempted to comprehensively review the current efficacy data, comparing WBI to PBI in terms of oncological outcome with special emphasis on local and systemic control. The analysis of the survival data has already been published [32]. The assessment of the adverse outcome data, including cosmesis and quality of life, will be reported separately.

## 2. Material and Methods

A systematic literature review in PubMed formed the basis of the analysis. This was carried out in accordance with the published PRISMA guideline and completed on 1 May 2021. In addition, we screened the major meetings (e.g., ASTRO, ESTRO, ESMO, and ASCO annual meetings) for published abstracts. The chosen keywords were “radiation therapy” or “radiotherapy” or “irradiation” AND “breast cancer” or “carcinoma of the breast” AND “partial” or “targeted” AND “randomized” OR “randomised” OR “randomly”. The inclusion criteria were randomized controlled trials including patients diagnosed with invasive breast cancer or carcinoma in situ comparing PBI to WBI. The trials were considered eligible when published after December 2009 in order to include comparable techniques and a homogeneous study population. 

To allow an estimation of the effect sizes comparing WBI to PBI, we extracted the published hazard and odds ratios as well as the event numbers from the identified trials. When no hazard ratios were reported, we estimated the hazard ratios and their corresponding 95% confidence intervals by reconstructing all events from the published survival curves or using the method published by Parmar and Tierney and colleagues [33,34]. When hazard ratios were neither reported nor estimable, we used the absolute number of events and calculated the odds ratio and the corresponding confidence interval. 

The aim of the present study was to evaluate the endpoints of ipsilateral breast tumor recurrence or local recurrence (IBTR), local recurrence-free survival (LRFS), local recurrences at the primary site (LRPS) and elsewhere in the ipsilateral breast (LREB), regional recurrences (RR), distant metastasis-free interval (DMFI), disease-free survival (DFS), contralateral breast cancer (CBC), and second primary cancer (SPC). The definition of disease-free survival was any first breast cancer-related event or death as defined in the included trials. For the analysis of the location of a local recurrence in the ipsilateral breast, we used the following definitions: recurrences at the primary site were those that originated within the margin of the tumor bed, and recurrences elsewhere were located in a different quadrant. The rates of cumulative incidence of a given endpoint were calculated for the full length of the available follow-up.

We used the inverse variance heterogeneity model (ivhet) to pool effect sizes, as this model uses a more conservative estimation of the confidence limits, produces lesser observed variances, and favors larger trials compared to the commonly used random effects model [35]. Zero-event correction was applied where appropriate [36]. The statistically significant limit was set at *p*-values lower than 0.05. Heterogeneity within the meta-analysis was obtained with Cochran’s Q-test with the corresponding *p*-values. Furthermore, we also described the I^2^ statistics where we defined values above 25% as considerable heterogeneity that triggered a subgroup analysis by the PBI technique as described in [37]. Funnel plots were created to assess publication bias. For statistical analysis, we used the Microsoft Excel add-in MetaXL 5.3 (EpiGear International, Sunrise Beach, Australia). Plots were created using Microsoft Excel for Microsoft Office 365 Pro Plus (Redmond, WA, USA). In order to obtain pooled event rates over the full course of follow-up or at the five-year time point, we calculated prevalence with the function embedded in MetaXL with the continuity correction set at 0.5. 

The analysis of available subgroups on IBTR was also performed when there were at least two trials reported on this patient group. In order to compare the effect of different radiation techniques, the effect of PBI vs. WBI was analyzed separately for trials using external beam radiation, intraoperative radiotherapy with electrons or photons, and for any form of brachytherapy. We acknowledge that this approach ignores the detailed differences between each individual techniques, which have their own characteristics. However, creating a subgroup for each technique used in the trials makes a general comparison impossible and ignores the basic approaches of each treatment. For the assessment of IBTR by the PBI technique described in Figure 2, we divided the results from the NSABP B-39 trial into the external beam technique and any form of brachytherapy.

When trials reported on the same endpoint in different publications, we attempted to include the most recent data in order to allow for the longest possible follow-up. For this reason, we have included the separate publications of the stratified prepathology and postpathology subgroups in the TARGIT trials [38,39,40,41].

## 3. Results

The results of the systematic literature search are presented in Figure A1, which revealed 40 publications reporting on thirteen different trials including a total number of 15,561 patients. Table 1 describes the details of the included studies with the respective inclusion criteria, treated patient cohorts, and interventions. In short, the trials included patients older than 40 years diagnosed with early-stage breast cancer with primary tumors up to 3 cm in size. Tumor biology consisted mainly of estrogen receptor-positive (83%) and node-negative disease (91.2%). Adjuvant endocrine therapy was the principal adjuvant systemic therapy (63.7%), whereas chemotherapy was applied less regularly (15.3%). A total of 16.8% of participants were younger than 50 years of age, and 9.8% had non-invasive tumors (DCIS). The median follow-up ranged between 2 and 17 years (median 8.6 years).

External beam radiotherapy (*n* = 9), intraoperative radiotherapy (IORT) (*n* = 3), and interstitial (single-entry catheter and multicatheter) brachytherapy (*n* = 3) were the PBI techniques used within the trials. The NSABP B-39 and the Budapest trials used EBRT and brachytherapy techniques. The different PBI schedules using EBRT were conventionally fractionated RT (*n* = 1) [42], once-daily hypofractionated RT (*n* = 5) [43,44,45,46,47], or twice-daily accelerated hypofractionated RT schedules (*n* = 4) [45,48,49,50]. The TARGIT-A trials used additional WBI in cases of prespecified risk factors in the PBI arms [38,39,40,41,51]. The funnel plots did not detect any publication bias. 

### 3.1. Local Control

First, we evaluated cumulative IBTR, as depicted in Figure 1 and Figure A2. The comparison showed a statistical difference between PBI and WBI when comparing odds ratios (OR = 1.66; CI-95%: 1.07–2.58; *p* = 0.024; I^2^ = 53.61) and pooled IBTR + LRFS (HR = 1.31; CI-95%: 0.96–1.78; *p* = 0.086). The comparison of the absolute cumulative incidence of IBTR in the PBI arms showed a difference of 1.35% (PBI rate: 3.4% vs. WBI rate: 2.05%) after median follow-up of 8.6 years (range 2–17 years). As we detected a significant heterogeneity in the analysis, we further examined the different PBI techniques. Here, we found a significant inferior local control rate for IORT (OR = 3.67; CI-95%: 2.28–5.90; *p* < 0.001). The other PBI techniques of EBRT/BT (OR = 1.25; CI-95%: 0.93–1.69; *p* = 0.146), EBRT (OR = 1.25; CI-95%: 0.85–1.84; *p* = 0.256), and BT (OR = 1.58; CI-95%: 0.52–4.73; *p* = 0.418) showed no differences in IBTR compared to WBI. After exclusion of the trials using IORT, the heterogeneity in the analysis was no longer present (*p* = 0.625).

An overview of the cumulative local recurrence rate over the published follow-up time in the different trials is shown in Figure A3.

Controlling for different lengths of follow-up, Table 2 shows the five-year IBTR rates separated by the PBI technique. Overall, at the five-year IBTR time point, PBI was not statistically different to WBI (2.47% vs. 1.46%; OR = 1.61 CI-95%: 0.97–2.66; *p* = 0.066). As shown, the analysis detected significant heterogeneity between the trials (*p* = 0.028). PBI using IORT showed significantly worse IBTR rates at five years (3.07% vs. 0.90%; OR = 3.39 CI-95%: 1.64–7.00; *p* = 0.001), comparing unfavorably to the other PBI techniques where no differences between PBI and WBI were detectable (EBRT/BT: 2.76% vs. 2.32%; EBRT: 1.7% vs. 1.41%; BT: 1.57% vs. 1.02%).

The following analysis, depicted in Figure 2, shows a comparison between PBI and WBI for all available data for the entire follow-up period. Here, the difference with Figure 1 is the splitting of the NSABP B-39 results separated by the PBI method. This modification still leads to higher pooled IBTR rates for PBI (OR = 1.71; CI-95%: 1.17–2.50; *p* = 0.005; I^2^ = 58.84) for all trials. In this analysis, IORT and all BT techniques (single catheter device, multicatheter technique) show higher statistically significant IBTR rates (IORT: HR = 3.67; CI-95%: 2.28–5.90; *p* < 0.001 and BT: HR = 2.03; CI-95%: 1.36–3.02; *p* = 0.001). The pooled raw numbers by technique showed a difference in cumulative IBTR rate between PBI and WBI for EBRT/BT: 1.65%; EBRT: 0.21%; BT: 0.62%; and IORT: 3.06%.

When separating IBTRs by location within the ipsilateral breast (Figure 3), we found no differences between PBI and WBI at the primary tumor site (LRPS: 1.36% vs. 1.34%; OR = 1.01; CI-95%: 0.65–1.59; *p* = 0.951; I^2^ = 32.68). The pooled estimate showed a higher IBTR rate for IORT (OR = 3.51; CI-95%: 1.36–9.11; *p* = 0.010). Recurrences elsewhere in the ipsilateral breast were more likely after treatment with PBI (LREB: 1.17% vs. 0.53%; OR = 2.21; CI-95%: 1.53–3.20; *p* < 0.001) (Figure 4) with a numerical difference of 0.64%.

According to Figure 5, tumor recurrences were equally distributed between LRPS and LREB in the trial arms applying PBI (OR = 1.00; CI-95%: 0.70–1.43; *p* = 0.986; I^2^ = 14.89). This is in contrast to the analysis in the WBI arms, where IBTRs were more likely at the primary site than elsewhere in the breast (OR = 2.20; CI-95%: 1.52–3.18; *p* < 0.001; I^2^ = 0).

### 3.2. Other Endpoints

Regional recurrences rates were higher in the PBI arms of the randomized trials than in the WBI arms as presented in Figure 6 (0.58% vs. 0.33%; OR = 1.75; CI-95%: 1.07–2.88; *p* < 0.001) with a numerical difference of 0.25%. Conversely, DMFI was not decreased in the PBI groups (97.2% vs. 97.4%; OR = 1.08; CI-95%: 0.89–1.30; *p* = 0.475) (Figure A4). The comparison shown in Figure A5 of DFS indicates significantly higher failure-free survival (OR = 1.16; CI-95%: 1.05–1.28; *p* = 0.003) after WBI. During follow-up, the calculated DFS rate was 86.4% after PBI and 88.3% after WBI, resulting in a numerical difference of 1.9%.

The incidence of contralateral breast cancer was not different between the groups (2% vs. 2.46%; OR = 0.81; CI-95%: 0.65–1.01; *p* = 0.067), as shown in Figure A6. The frequency of second primary cancers (Figure A7) others than breast cancers was equally distributed between PBI and WBI (5.51% vs. 5.04%; OR = 1.09; CI-95%: 0.85–1.40; *p* = 0.481).

Table 3 compares the anticipated absolute effects based on the relative effects and the risk per 100 for PBI and WBI. 

The subgroup analysis of IBTR is shown in Table 4. Because of the detected heterogeneity in the IBTR analysis, this assessment was influenced by the different trials and PBI methods. None of the investigated subgroups showed a significant effect on the comparison of PBI vs. WBI, as demonstrated by the non-significant interaction tests. Statistically inferior IBTR in the PBI arms was detected in tumors of a size between 11 and 20 mm (HR = 2.53; CI-95%: 1.22–5.28; *p* = 0.013), primary tumor of a size greater than 1.5 cm (HR = 2.71; CI-95%: 1.28–5.72; *p* = 0.009), N1 disease (HR = 2.82; CI-95%: 1.41–5.62; *p* = 0.003), and Her2-negative status (HR = 3.92; CI-95%: 1.15–13.40; *p* = 0.029). 

The statistical comparison did not find any differences in IBTR in the subgroups of DCIS; hormone receptor-negative cancers; grade III, high-risk criteria according to ASTRO consensus; or younger women under the age of 50 years. 

## 4. Discussion

### 4.1. Main Results

Numerous randomized trials using various techniques of partial-breast irradiation have attempted to demonstrate non-inferior local control rates in early-stage breast cancer. Some were able to confirm this hypothesis [39,41,43,49,50,60,69,78], whereas others could not conclude non-inferiority between PBI and WBI [38,62]. Our meta-analysis shows that the pooled PBI trials using different techniques are formally inferior in local control rates compared to whole-breast irradiation with a statistically significant numerical difference of 1.35% after a follow-up ranging between 3 and 17 years. After a period of 5 years, the numerical difference was 1.01%. Additionally, we found strong suggestions that selecting patients according to risk group and utilizing a suitable method influence the efficacy of PBI. When analyzed by the location in the breast, the treatments were similarly effective at the primary site (LRPS rates: 1.47% vs. 1.34%), showing that the remaining microscopic disease at the tumor bed is adequately treated by PBI. However, PBI led to more recurrences elsewhere in the breast, with raw incidences of 1.26% after PBI compared to 0.53% after WBI. This finding demonstrates that WBI is superior in treating microscopic tumor foci that are distant from the resected primary tumor. Furthermore, this confirms the past analysis of the START-trials, where WBI reduced the rate of ipsilateral new primary tumors [81]. 

The RAPID investigators first reported on differences in the IBTR distribution within the ipsilateral breast after PBI and WBI [49]. In the present meta-analysis. we can confirm this observation with the ratio of primary site vs. other ipsilateral recurrences of 1.5% to 1.4% (~1:1) after PBI and 1.4% to 0.6% (~2.3:1) after WBI. An explanation for this phenomenon could be that other foci beyond the original tumor site were controlled with WBI, while the cells at the original site might be more resistant due to higher tumor cell density, more hypoxia, or other post-therapeutic changes.

The results from several prospective randomized trials demonstrated that in-breast failures are distributed approximately 75% to the primary site and 25% to elsewhere in the surrounding tissue [25,26,27,28,82]. In this analysis, we found a similar ratio in the patients receiving WBI. In contrast, recurrences after PBI were equally distributed across the breast. Ipsilateral breast recurrences can originate from cells located far from the original tumor bed and occur at a clinically and statistically detectable rate. Earlier opinions questioned whether PBI could reach non-inferior efficacy, because with an assumed 10% local failure rate after 10-year follow-up, the proportion of other in-breast recurrences would reach around 2.5%. However, in this report, the cumulative incidence of ipsilateral local failures outside the tumor bed region was only 0.6–1.4%, mirroring the rapid advances in breast cancer treatment.

The detectable increase in regional nodal recurrences rates after PBI reinforces the role of the incidental radiation dose in the lower axillary levels or the internal mammary nodes. The incidental dose applied to the axillary lymph nodes during whole-breast radiotherapy has been postulated to affect axillary relapses since the publication of the ACOSOG Z0011 trial [83,84]. The recently published dosimetric data from the prospective randomized INSEMA trial showed that at least 25–50% of the patients received an unintended therapeutic dose to the axillary level I lymph nodes [85]. Whether regional failures occurred more frequently in patients with positive lymph nodes or a higher risk profile is unknown.

The absolute increase in local and regional relapses with PBI was small (0.3–0.8%), which questions the clinical relevance of this observation. It is reassuring that PBI did not affect systemic control, as shown by the similar occurrence of distant metastases of both treatments. This is in accordance with our recently published assessment of mortality, which confirmed no differences in overall, breast cancer-specific-, and non-breast cancer mortality between PBI and WBI [32].

The presumed lower doses to organs at risk currently did not translate into a reduction in the total number of cases of contralateral or second non-breast cancers. In the analysis of the EBCTCG data by Taylor and colleagues, the contralateral breast cancer risk associated with WBI was detectable with more than five years of fol-low-up, while the incidence of lung cancers started becoming evident only after more than 10 years [86]. Longer follow-up will have to determine whether PBI can reduce the risk of second cancers.

The rate at which patients relapse elsewhere in the breast or regionally should be similar to the risk factors for a CBC, which typically include biological risk groups, such as lobular histology, lack of an estrogen receptor, a higher proliferation index, or a history of receipt of radiation therapy. Conversely, systemic therapies, such as endocrine-, chemo-, and Her2-targeted therapies are protective agents [87,88]. Therefore, the logical conclusion is to select low-risk patients for partial-breast treatments where the risk of relapse in the remaining glandular tissue and regional lymph nodes is either negligible or is adequately treated by adjuvant systemic therapy [62,89]. The role of systemic agents in controlling residual disease elsewhere in the breast instead of WBI will be interesting, as we know that endocrine therapy alone cannot substitute for WBI in controlling disease at the primary site [16]. Hopefully, future investigations will further examine PBI in subgroups receiving adjuvant systemic therapies or no adjuvant therapy. Especially in Her2-positive patients, the use of very effective systemic treatments might lower the need for radiation therapy to treat the microscopic remnant cancer cells.

### 4.2. Techniques

Our analysis of IBTR revealed a significant heterogeneity in the comparison, which might be attributable to either difference in risk groups in the selected patients or PBI techniques. The analysis by PBI methods suggests that PBI by EBRT achieved similar local control, while techniques such as IORT and BT may be associated with higher recurrence rates. In the included trials, BT was performed as a multicatheter approach (NSABP B-39, GEC-ESTRO, Budapest) or a single-catheter treatment (NSABP B-39). The long-term analysis of the NSABP B-39 trial suggested that the failure of achieving equivalence in IBTR was driven by the patients treated with BT. Notably, the majority of these patients (~80%) were treated with single-device brachytherapy, which is a highly debated approach due to its concentric rigid dose distribution. The largest registry series reported an actuarial LR rate of 3.8% after five years, which compares unfavorable to other PBI techniques [90]. However, both BT techniques reported similar effect sizes compared to that of WBI (HR multicatheter: 2.21 CI-95%: 1.10–4.46; HR single-catheter: 2.15 CI-95%: 1.34–3.44; 10 y cumulative incidence: 7.7% and 7.8%, respectively). Furthermore, the primary endpoint analysis was not stratified by the irradiation technique, and the subgroup analysis was reported per protocol, which means that imbalances in the subgroups could also explain this finding. In contrast, the GEC-ESTRO trial reported equivalent IBTR with multicatheter BT to WBI. This difference could be due to a shorter follow-up period, a higher level of expertise of the treating centers, the sole use of multicatheter BT, or the inclusion of lower risk patients in the GEC-ESTRO trial.

Similarly, trials with PBI using the IORT technique also showed inferior local control rates. This was also the case when only low-risk patients (ELIOT trial) were selected or IORT was restricted to immediate IORT during lumpectomy only (TARGIT prepath.) compared to other techniques. There are several details to be considered here. The TARGIT trials included patients treated with IORT during the initial lumpectomy (prepathology) or during a second surgery where the initial scar was reopened (postpathology). The different approaches introduced considerable differences in the risk groups between the strata. It should also be noted that the TARGIT-A trial did not purely test partial- vs. whole-breast radiotherapy but included a risk-adapted approach of IORT with additional WBI in the presence of certain risk factors, resulting in a proportion of 26.8% in the prepathology and 5.7% in the postpathology stratum receiving both treatments [38,41]. We were unable to analyze the long-term local control rate in the TARGIT trials beyond five years, as the numbers were reported as the pooled endpoint of LRFS. Moreover, the technical differences of both IORT techniques using photon and electron irradiation must also be considered. Electron treatment covers larger intraoperative volumes with a homogenous dose distribution, while KV-based IORT applies heterogeneously distributed energy.

It is unclear whether the observed tendencies of a higher IBTR risk for IORT and BT are due to a higher risk of missing residual tumor cells at the tumor bed or an inclusion of higher risk patients. The analysis of the GEC-ESTRO trial does not indicate any evidence of missing microscopic disease at the tumor bed, as equal numbers of IBTRs at the primary site were reported for PBI and WBI. This is in contrast to the trials using IORT, were we detected higher recurrence rates at the primary tumor site (Figure 3).

### 4.3. Subgroups

At present, the leading radiation oncology societies recommend the use of partial-breast radiation in selected low-risk patients [91,92]. The presented subgroup analysis was unable to identify a specific subgroup unsuitable for PBI given by the non-significant interaction tests. However, as we detected heterogeneity in the included trials in the main analysis, the assessment of the different subgroups might be biased by the PBI techniques.

Nonetheless, multiple exploratory subgroups showed numerically higher recurrence rates when using PBI in patients with known risk factors for a local relapse, such as tumor size and nodal positivity. Therefore, it is difficult to make firm conclusions, as the sample size and different techniques limited this analysis. We await the planned individual patient data analysis by the early breast cancer trialist collaborative group (EBCTCG).

According to the ASTRO consensus, DCIS is included in the “suitable group” when the lesion was screening detected, had low to intermediate nuclear grade and a size of ≤2.5 cm, and was resected with negative margins ≥ 3 mm [92,93], as multiple prospective single arm trials reported favorable outcomes [94,95,96]. The appropriateness of PBI in women with DCIS has not been generally established, as randomized trials included only a small number of patients with non-invasive tumors, because the pattern of recurrence as well as the spread within the breast might differ to those in invasive cancers [60,69]. The NSABP B-39 and RAPID trials were the first to include a larger sample of DCIS patients. The presented subgroup analysis strengthens the argument to expand the use of PBI, as we showed no differences in IBTR between 1412 prerandomization-stratified patients (HR = 1.21; CI-95%: 0.69–2.11).

When comparing different techniques of partial-breast RT, one has to acknowledge that clinical target volume and dose application differ substantially. IORT and brachytherapy deliver (by nature of the technique) an inhomogeneous dose within the target volume. The TARGIT applicator reaches single-fraction surface doses from 20 to 31 Gy with a reduction to 4–8 Gy at a depth of 1 cm [41,97,98]. In contrast, IORT using electrons delivers a more homogenously distributed dose over a larger area. The IMPORT LOW trial applied EBRT using “mini-tangents” with much larger treated volumes and a homogenous dose distribution. Furthermore, the definition of a clinical target volume is complex and has to account for the surgical technique, including oncoplastic methods, the preoperative tumor location, the surgical path and scar, and the RT technique. The ideal target volume is concentric around the original tumor, which usually does not correlate well with the surgical tumor cavity since the surgical resection defect is rarely uniformly around the tumor. When comparing the absolute numbers, it is notable that the only techniques that reduced local failures are the partial tangents used in the IMPORT LOW trial [43].

The IBTR rates after PBI have to be weighed against the number reported by studies testing the omission of radiotherapy in low-risk breast cancer. The randomized PRIME II trial calculated an actuarial 5- and 10-year local control of ET alone of 4.1% and 9.8% compared to 1.3% and 0.9%, respectively, after ET and WBI [8,99]. Despite the higher risk population included in the PBI trials, we did not observe local recurrence rates reaching 10% after 10 years follow-up, demonstrating that the indirect comparison between partial-breast treatment combined with systemic therapy and endocrine therapy alone suggests that the former is superior.

### 4.4. Limitations of Analysis

Our analysis is based on pooled, published data that does not regard individual patients, which would be generally preferable and might allow a more comprehensive subgroup analysis. The fractionation schedules in the control arm using WBI where both normofractionated RT (fraction sizes between 1.8 and 2 Gy per fraction) and hypofractionated RT (>2.5 Gy per fraction). In some trials, hypofractionated schedules were associated with a reduction in IBTR; this might have introduced heterogeneity into the control arms of this analysis, which we could not statistically control [100]. Using an alpha/beta ratio of 3.7 for breast cancer [101], the dose comparison of some WBI regimes results in a higher dose compared to that of most of the PBI regimes included in this analysis. Additionally, many trials applied tumor bed boosts to their WBI arms, which are known to further reduce the local recurrence rates [75,102,103]. The similar failure rate at the primary tumor site questions the use of a tumor bed boost (dose escalation) in the included trial population. Another factor adding heterogeneity is the different follow-up times, which range from 2 to 17 years. However, the majority of the studies reported on an observation period of at least five years, which is usually considered long enough to assess the effect of radiotherapy on local recurrences [26,104].

Numerous meta-analyses and systematic reviews have been published on this subject since 2010 [105,106,107,108,109,110,111,112,113,114]. The number of included patients and follow-up time limited earlier analyses [105,106,107,108,109,110,111]. Newer publications either lack the inclusion of larger recently published trials [112] or excluded the three randomized trials performing PBI with IORT [114]. The report published by Viani and colleagues reported similar results to ours; however, a strength of our analysis is the detailed assessment of local relapse with localization and distribution in addition to a high number of included patients (*n* = 15,661) [113].

## 5. Conclusions

Partial-breast radiotherapy achieves equivalent oncological outcomes to those of whole-breast radiotherapy when selecting low-risk patients and using appropriate techniques. The appropriateness of limiting the target volume to a partial treatment of the breast depends on the individual risk profile and the margin of acceptance for non-inferiority.

## Figures and Tables

**Figure 1 cancers-13-02967-f001:**
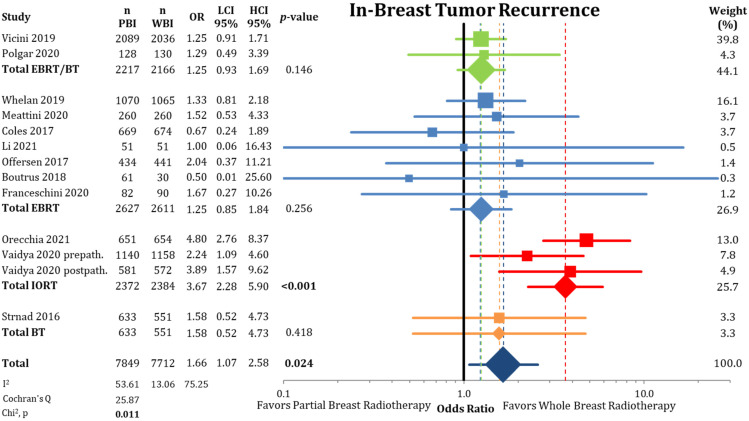
Comparison of in-breast tumor recurrences between partial-breast radiotherapy and whole-breast radiotherapy. The odds ratios for each trial grouped by radiation technique and the pooled effect sizes with corresponding 95% confidence intervals are reported. The squares represent the effect sizes of the individual trials, while the center of the diamonds indicate the pooled odds ratios for the individual techniques and the overall effect. Heterogeneity analysis is shown using the I^2^ with 95% confidence intervals and Cochran’s Q analysis. Bold *p*-values indicate statistically significant results.

**Figure 2 cancers-13-02967-f002:**
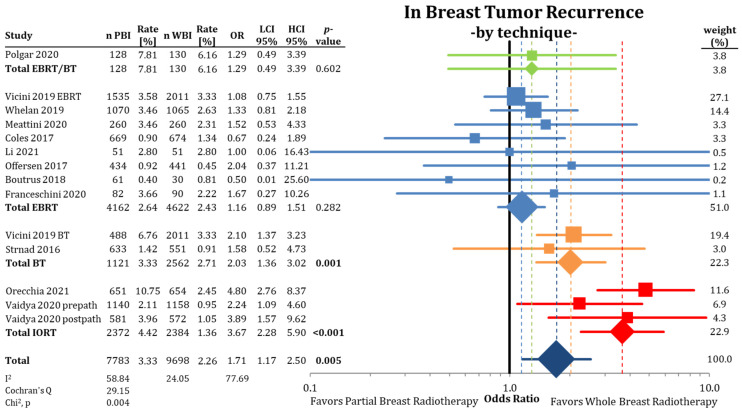
Analysis of in-breast tumor recurrence by radiation technique using odds ratios with a 95% confidence interval. The squares represent the effect sizes of the individual trials, while the center of the diamonds indicate the pooled odds ratios for the individual techniques and the overall effect. Because of the splitting of the NSABP B-39 trial [53], the WBI arm is counted twice in the comparison. The pooled estimate should be interpreted accordingly. Bold *p*-values are statistically significant.

**Figure 3 cancers-13-02967-f003:**
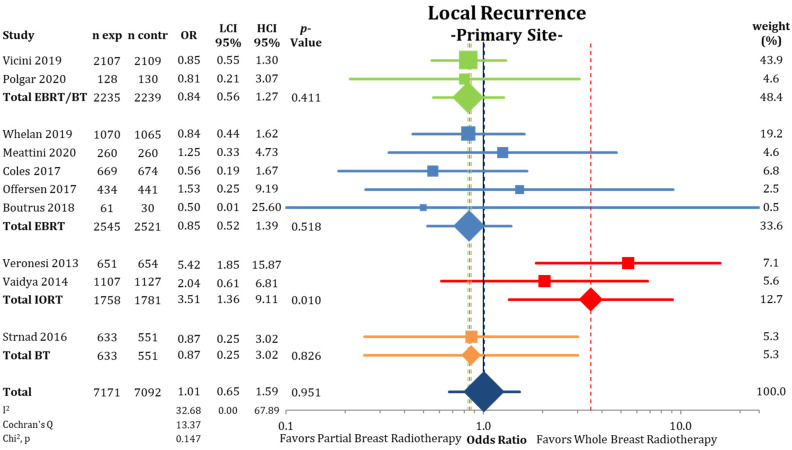
Evaluation of the local recurrences located at the primary site or marginally to the primary tumor using odds ratios with squares representing individual trials and diamonds the pooled effect sizes for technique and overall analysis.

**Figure 4 cancers-13-02967-f004:**
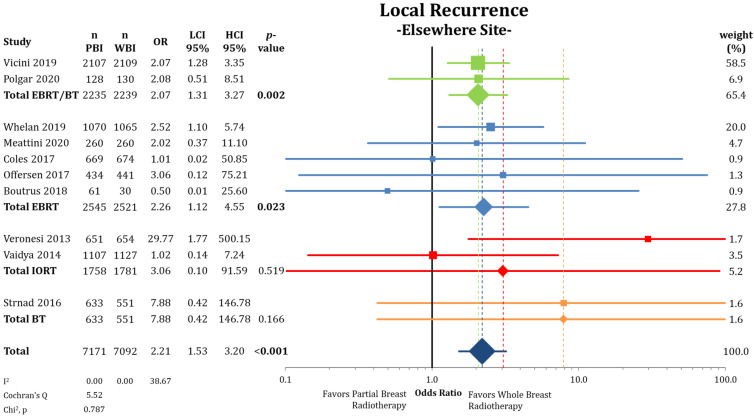
Evaluation of the local recurrences elsewhere in the ipsilateral breast using odds ratios with squares representing indi-vidual trials and diamonds the pooled effect sizes for technique and overall analysis. Bold *p*-values indicate statistically significant results.

**Figure 5 cancers-13-02967-f005:**
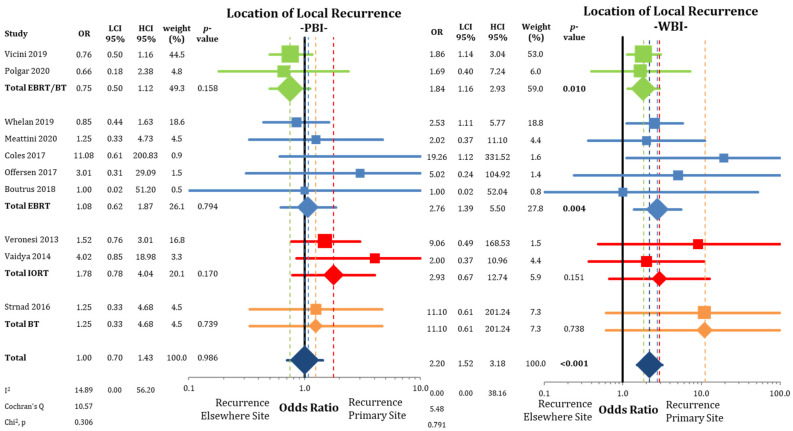
Assessment of the location of the local recurrences (in-breast tumor recurrences) after partial-breast irradiation and whole-breast irradiation with the use of odds ratios. Squares represent individual trials and diamonds the pooled effect sizes for technique and overall analysis. Bold *p*-values indicate statistically significant results.

**Figure 6 cancers-13-02967-f006:**
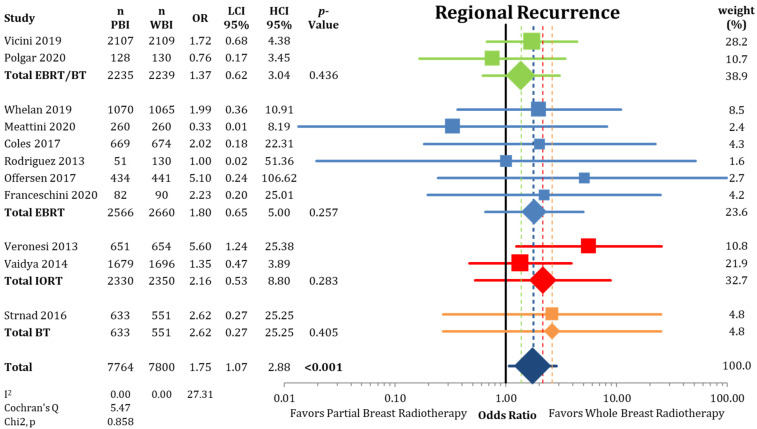
Assessment of regional recurrences between partial-breast radiotherapy and whole-breast radiotherapy. The odds ratios for each trial (squares) grouped by radiation technique and the pooled effect sizes (diamonds) with corresponding 95% confidence intervals are displayed. Bold *p*-values indicate statistically significant results.

**Table 1 cancers-13-02967-t001:** Overview of the included trials with relevant patient characteristics.

Study	Synonym	Additional Publications	Years of Trial	FU	NTotal	Med. Age	Stat. Setting	Prim. EP	Population	Stratification Factors	PBI Technique	PBI Dose	WBI Dose	G3	DCIS	N+	HR+	Her2+	CTx	ET	Boost
Vicini 2019 [48]	NSABP B-39	Vicini et al. 2019 [52] White et al. 2019 [53]	2005–2013	10.2	4216	54	Equiv.	IBTR	IBC or DCIS; T < 3 cm, ≤N1; R0; >18 y	Stage, menopausal, ER, CTx	3DCRT, single- and multicath. BT	34/3.4; 38.5/3.85 10x in 5–8 d	50/2; 50.4/1.8; opt. Boost	26%	24%	10%	81%	n.r.	29%	n.r.	80%
Whelan 2019 [49]	RAPID	Olivotto et al. 2013 [54] Peterson et al. 2015 [55] Whelan et al. 2019 [56]	02/2006–07/2011	8.6	2135	61	noninf.	IBTR	IBC or DCIS; T < 3 cm; R0; N0; >40 y; unifocal	Age >< 50; histology, T >< 1.5 cm; ER, center	3DCRT IMRT	38.5/3.85 BID in 5–8 d	50/2; 42.5/2.66+ opt. Boost	16%	18%	0%	84%	6%	13%	55%	21%
Meattini 2020 [46]	Florence	Livi et al. 2010 [57] Livi et al. 2015 [58] Meattini et al. 2017 [59] Meattini et al. 2020 [60]	03/2005–06/2013	10.7	520	n.r.	Equiv.	IBTR	IBC or DCIS; T < 2.5 cm; >40 y; BCS+	None	IMRT	30/6 q.o.d	50/2+ opt. Boost 10/2	11.4%	11%	10%	96%	4%	7%	62%	n.r.
Orecchia 2021 [61]	ELIOT	Veronesi et al. 2013 [62]	11/2000–12/2007	12.4	1305	n.r.	Equiv.	IBTR	IBC; T < 2.5 cm; cN0, R0; 48–75 y; unifocal	T < 1 cm, T 1–1.4 cm, T > 1.5 cm	IORT e-	21/21	50/2+ opt. Boost 10/2	20.9%	0%	27%	91%	3%	22%	89%	n.r.
Vaidya 2020 [41]	TARGIT-A prepathology	Vaidya et al. 2010 [51] Andersen et al. 2012 [63] Sperk et al. 2012 [64] Welzel et al. 2013 [65] Keshtgar et al. 2013 [66] Vaidya 2014 [39] Corica et al. 2016 [67] Corica et al. 2018 [68]	03/2000–06/2012	8.6	2298	Mean 63	noninf.	LRFS	IDC; T < 2.5 cm; R0; >45 y; unifocal	Center, timing	IORT x	20/20	n.r.	20%	0%	~21%	90%	15%	21%	81%	38%
Vaidya 2020 [38]	TARGIT-A postpathology		03/2000–06/2012	9	1153	Mean 63	noninf.	LRFS	IDC; T < 2.5 cm; R0; >45 y; unifocal	Center, timing	IORT x	20/20	n.r.	6%	3%	5%	98%	6%	4%	87%	n.r.
Strnad 2016 [69]	GEC Estro	Polgar et al. 2017 [70] Schäfer et al. 2018 [71]	04/2004–07/2009	6.6	1328	62	noninf.	IBTR	IBC or DCIS; T < 3 cm; R0; N0; >40 y; BCS+	Center, menopausal, stage	multicath. BT	32/4; 30.3/4.3 or PDR	50/2; 50.4/1.8; Boost opt.	8.3%	5%	6%	95%	n.r.	11%	90%	98%
Coles 2017 [43]	Import low	Bhattacharya et al. 2019 [72] Bhattacharya et al. 2019 [73] Bhattacharya et al. 2019 [74]	05/2007–10/2010	6	1343	62	noninf.	IBTR	IDC; T < 3 cm; >50 y; pN0–1	Centre	3DCRT	40/2.67 QD	40/2.67	9.7%	0%	3%	95%	4%	5%	80%	n.r.
Polgar 2020 [42]	Budapest	Polgar et al. 2004 [75] Polgar et al. 2007 [76] Lövey et al. 2007 [77] Polgar et al. 2013 [78] Polgar et al. 2014 [79]	1998–2004	17	258	Mean 59	noninf.	LR	IBC; T < 2 cm; N0; R0; G1–2; unifocal	None	multicath. BT 3DCRT e-	BT: 36.4/5.2 BID; e-:50/2 QD	50/2+ opt. 16/2	0.0%	0%	0%	88%	n.r.	3%	99%	0.8%
Li 2021 [80]	Barcelona	Rodriguez et al. 2013 [50]	2007–2013	10.3	102	Mean 68	noninf.	IBTR	IBC; T < 3 cm; R0; N0; >60 y; unifocal; G 1–2	n.r.	3DCRT	37.5/3.75 BID	48/2+ opt. Boost 10/2 or 20/2	0.0%	0%	0%	98%	1%	3%	99%	n.r.
Offersen 2017 [44]	DBCG PBI		2009–2016	3	882	66	noninf.	Breast induration	IBC, T1, R0, >60 y, G1–2, HER2−, pN0	Center, ET	3DCRT	40/2.67 QD	40/2.67	<1.0%	0%	0%	100%	0%	n.r.	80%	n.r.
Boutrus 2018 [45]	Cairo		n.r.	2	91	50	n.r.	IBTR	IBC; T < 3 cm; R0; N0; >40 y; unifocal; G1–3	n.r.	3DCRT	38.5/3.85 QD 38.5/3.85 BID	50/2+ opt: Boost	n.r.	0%	0%	80%	n.r.	19%	n.r.	n.r.
Franceschini 2020 [47]	HYPAB		01/2015–01/2018	3	172	64		Cosmesis	T1–2, postmeno, cN0, BCS, ER +, unicentric, R0 > 5 mm,	n.r.	VMAT	30/6 q.o.d	40.5/2.7 SIB to 48/3.2	3%	0%	n.r.	100%	n.r.	n.r.	97%	100%

**Table 2 cancers-13-02967-t002:** Analysis of in-breast tumor recurrences after a five-year follow-up period between partial-breast irradiation and whole-breast irradiation for the whole group and by technique. The prevalence rates by treatment arm and the statistical comparison using the odds ratios with the corresponding 95% confidence intervals are given. The bold *p*-values are considered statistically significant.

Five-Year in-Breast Tumor Recurrence							
Study	Rate [%] PBI	Rate [%] WBI	OR	LCI 95%	HCI 95%	Weight (%)	*p*-Value
NSABP B-39	2.68	2.26	1.19	0.80	1.77	42.5	
Budapest	3.91	3.08	1.28	0.34	4.88	3.7	
IBTR 5y EBTR/BT	2.76	2.32	1.20	0.82	1.75	46.2	0.349
RAPID	2.35	1.67	1.42	0.75	2.70	15.9	
IMPORT LOW	0.53	1.04	0.51	0.13	1.97	3.6	
Florence	2.31	1.15	2.02	0.50	8.18	3.4	
IBTR 5y EBTR	1.70	1.41	1.17	0.44	3.15	22.9	0.749
ELIOT	4.26	0.52	8.53	2.56	28.50	4.6	
TARGIT prepath	2.11	0.95	2.24	1.09	4.60	12.8	
TARGIT postpath	3.96	1.05	3.89	1.57	9.62	8.1	
IBTR 5y IORT	3.07	0.90	3.39	1.64	7.00	25.4	**0.001**
GEC ESTRO	1.57	1.02	1.55	0.52	4.65	5.5	
IBTR 5y BT	1.57	1.02	1.55	0.52	4.65	5.5	0.436
Total 5y IBTR	2.47	1.46	1.61	0.97	2.66	100.0	0.066
I^2^			53.45	1.14	78.08		
Cochran’s Q			17.19				
Chi^2^, *p*			**0.028**				

**Table 3 cancers-13-02967-t003:** Overview of investigated endpoints comparing partial- to whole-breast radiotherapy and showing the relative effect (odds ratio) and the anticipated absolute effects per 100 patients.

Outcome (Median Follow-Up Range)	Anticipated Absolute Effects				Relative Effect (95% CI)			No. of Participants (Studies)
	Risk with WBI per 100	Risk with PBI (95% CI) per 100						
In-breast tumor recurrence (2–17 years)	2.05	3.40	2.19	5.27	1.66	1.07	2.58	15,561 (13 RCTs)
In-breast tumor recurrence at primary site (2–17 years)	1.34	1.36	0.87	2.13	1.01	0.65	1.59	14,161 (10 RCTs)
In-breast tumor recurrence at elsewhere site (2–17 years)	0.53	1.17	0.81	1.69	2.21	1.53	3.20	14,161 (10 RCTs)
Regional recurrence (3–17 years)	0.33	0.58	0.35	0.95	1.75	1.07	2.88	15,485 (11 RCTs)
Distant metastasis-free interval (3–17 years)	2.60	2.80	2.31	3.39	1.08	0.89	1.30	15,222 (10 RCTs)
Disease-free survival (3–17 years)	11.71	13.60	12.34	15.00	1.16	1.05	1.28	14,778 (9 RCTs)
Contralateral breast cancer (3–12.4 years)	2.58	2.10	1.68	2.62	0.81	0.65	1.01	13,473 (9 RCTs)
Second primary cancer (3–12.4 years)	5.04	5.51	4.30	7.06	1.09	0.85	1.40	11,745 (8 RCTs)

**Table 4 cancers-13-02967-t004:** Subgroup comparison of PBI and WBI regarding in-breast tumor recurrences. Hazard ratios and the corresponding 95% confidence intervals as well as the *p*-values for the interaction test of the subgroups are reported. Hazard ratios below 1 favor PBI and those above 1 favor WBI. Bold *p*-values represent statistically significant results.

Study	HR	LCI 95%	HCI 95%	Weight (%)	*p*-Value	Study	HR	LCI 95%	HCI 95%	Weight (%)	*p*-Value
Age < 50 years						N0					
Whelan 2019 [49]	0.78	0.29	2.11	72.6		Vicini 2019 [48]	1.31	0.85	2.00	40.1	
Livi 2015 [58]	1.52	0.96	24.23	27.4		Whelan 2019 [49]	1.27	0.84	1.91	43.6	
Total	0.94	0.40	2.18	100.0	0.879	Livi 2015 [58]	1.08	0.15	7.70	1.9	
Age > 50 years						Orecchia 2021 [61]	5.47	2.68	11.19	14.4	
Whelan 2019 [49]	1.44	0.91	2.11	84.2		Total	1.58	0.76	3.28	100	0.219
Livi 2015 [58]	1.07	0.07	17.08	2.0		N1					
Coles 2017 [43]	0.65	0.23	1.84	13.8		Vicini 2019 [48]	1.91	0.57	6.34	32.9	
Total	1.28	0.87	1.89	100.0	0.206	Orecchia 2021 [61]	3.41	1.47	7.92	67.1	
Age > 70 years						Total	2.82	1.41	5.62	100.0	**0.003**
Livi 2015 [58]	1.07	0.07	17.08	23.1		Interaction N0/N1				0.260	
Orecchia 2021 [61]	1.86	0.41	8.33	76.9		Grade 1–2					
Total	1.64	0.44	6.13	100.0	0.464	Whelan 2019 [49]	1.10	0.60	2.01	54.2	
Interaction Age					0.732	Orecchia 2021 [61]	4.50	2.33	8.68	45.8	
DCIS						Total	2.10	0.53	8.37	100.0	0.294
Vicini 2019 [48]	1.01	0.61	1.68	69.7		Grade 3					
Whelan 2019 [49]	1.81	0.84	3.91	30.3		Whelan 2019 [49]	1.06	0.44	2.55	42.4	
Total	1.21	0.69	2.11	100.0	0.513	Livi 2015 [58]	1.43	0.09	22.92	4.3	
Invasive Cancer						Orecchia 2021 [61]	2.18	1.00	4.79	53.3	
Vicini 2019 [48]	1.37	0.91	2.05	24.3		Total	1.58	0.89	2.79	100.0	0.118
Whelan 2019 [49]	1.12	0.69	1.84	16.4		Interaction G1–2/G3				0.709	
Orecchia 2021 [61]	4.62	2.68	7.95	13.4		Estrogen Receptor Positive					
Vaidya 2014 [39]	2.13	1.01	4.49	7.1		Vicini 2019 [48]	1.32	0.91	1.92	50.5	
Coles 2017 [43]	0.65	0.23	1.84	3.7		Whelan 2019 [49]	1.19	0.69	2.07	23.3	
Polgar 2013 [78]	1.09	0.88	1.72	35.2		Livi 2015 [58]	1.79	0.80	10.69	4.2	
Total	1.44	0.85	2.44	100.0	0.171	Orecchia 2021 [61]	4.21	2.39	7.42	21.9	
Interaction DCIS/Invasive					0.644	Total	1.68	0.84	3.35	100.0	0.139
T1a/b						Estrogen Receptor Negative					
Vicini 2019 [48]	0.58	0.27	1.22	68.2		Vicini 2019 [48]	0.98	0.54	1.77	72.6	
Orecchia 2021 [61]	4.01	1.33	12.1	31.8		Whelan 2019 [49]	1.01	0.34	3.04	21.3	
Total	1.07	0.15	7.82	100.0	0.945	Orecchia 2021 [61]	9.25	1.19	71.70	6.1	
T1c						Total	1.13	0.40	3.16	100.0	0.815
Vicini 2019 [48]	2.66	1.24	5.68	48.2		Interaction ER + /ER-				0.529	
Livi 2015 [58]	1.32	0.08	21.23	3.6		Her2 Negative					
Orecchia 2021 [61]	4.91	2.30	10.51	48.3		Orecchia 2021 [61]	4.35	2.47	7.64	92.3	
Total	2.53	1.22	5.28	100.0	**0.013**	Livi 2015 [58]	1.13	0.16	8.02	7.7	
T2						Total	3.92	1.15	13.40	100.0	**0.029**
Vicini 2019 [48]	1.34	0.52	3.46	57.9		ASTRO Risk Group Criteria—Suitable					
Orecchia 2021 [61]	4.80	1.58	14.58	42.1		Vicini 2019 [48]	1.12	0.46	2.76	32.0	
Total	2.29	0.65	8.05	100.0	0.195	Whelan 2019 [49]	1.06	0.57	1.95	68.0	
Interaction T1ab/T1c/T2					0.729	Total	1.08	0.65	1.79	100.0	0.769
T < 1.5 cm						ASTRO Risk Group Criteria—Not Suitable					
Whelan 2019 [49]	1.02	0.59	1.75	63.7		Vicini 2019 [48]	1.26	0.77	2.08	56.6	
Orecchia 2021 [61]	4.92	2.39	10.12	36.3		Whelan 2019 [49]	1.46	0.83	2.58	43.4	
Total	1.80	0.37	8.85	100.0	0.467	Total	1.34	0.92	1.95	100.0	0.122
T > 1.5 cm						Interaction ASTRO Risk Group					0.495
Whelan 2019 [49]	2.01	1.03	3.93	60.8							
Orecchia 2021 [61]	4.31	1.87	9.92	39.2							
Total	2.71	1.28	5.72	100.0	**0.009**						
Interaction T-size 1.5 cm					0.650						

## Data Availability

All data used in the analysis are available from the cited published trials.

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
