# Peer review of "Comparing Local and Systemic Control between Partial- and Whole-Breast Radiotherapy in Low-Risk Breast Cancer—A Meta-Analysis of Randomized Trials"

_cancers, 2021, doi:10.3390/cancers13122967_

Round 1
Reviewer 1 Report
The article is a meta-analysis of randomized trials comparing whole breast irradiation and partial breast irradiation, utilizing randomized trials published after 2009. It is well written, well organized, and provides useful clinical data. The methods and statistical analysis is well-described. References are comprehensive. 40 publications were included, describing 13 studies that met inclusion criteria. Median f/u on the included trials is 8.6 years, which is good. This paper is part of the 3 manuscript series by the same authors, one already published on overall survival, another in preparation regarding cosmetic outcomes.
This is not the first meta-analysis published on this topic. Review of pubmed reveals a dozen such meta-analyses published since 2010. The authors mention one of these, but should include a more comprehensive discussion of these prior studies, and describe how their analysis differs and contributes beyond that already published. The current study does include a more modern collection of patients, presumably with more consistent techniques, and there statistical analysis does seem more sophisticated that in some of the other publications.
A few other suggestions:
1) page 2, line 68: change recurrences to recurrence.
2) page 13, line 333: change "Notable" to "Notably, "
3) page 14, line 402: the numbers presented appear to be local recurrence rates, not local control rates as stated.
Author Response
Dear editor,
We would to thank both reviewers for their thorough, helpful and constructive critic of our manuscript. We have addressed all comments and embedded them into our manuscript. Further, we provided a point-by-point reply below.
Best wishes,
Edwin Bölke, Jan Haussmann, Christiane Matuschek
Reviewer 1
The article is a meta-analysis of randomized trials comparing whole breast irradiation and partial breast irradiation, utilizing randomized trials published after 2009. It is well written, well organized, and provides useful clinical data. The methods and statistical analysis is well-described. References are comprehensive. 40 publications were included, describing 13 studies that met inclusion criteria. Median f/u on the included trials is 8.6 years, which is good. This paper is part of the 3 manuscript series by the same authors, one already published on overall survival, another in preparation regarding cosmetic outcomes.
This is not the first meta-analysis published on this topic. Review of pubmed reveals a dozen such meta-analyses published since 2010. The authors mention one of these, but should include a more comprehensive discussion of these prior studies, and describe how their analysis differs and contributes beyond that already published. The current study does include a more modern collection of patients, presumably with more consistent techniques, and there statistical analysis does seem more sophisticated that in some of the other publications.
We have added the references of other meta-analyses and pointed out some differences to other works.
A few other suggestions:
1) page 2, line 68: change recurrences to recurrence.
We have changed the line accordingly.
2) page 13, line 333: change "Notable" to "Notably, "
We have changed the line accordingly.
3) page 14, line 402: the numbers presented appear to be local recurrence rates, not local control rates as stated.
We have changed the line accordingly.

Reviewer 2 Report
Dear authors,
overall, your manuscript is well written, the analysis is sound, conclusions are adequate, and the topic is interesting. Nevertheless, there are several points that should be improved.
Abstract
- "IBTR was significantly different between.." Please state the direction of the difference.
- The conclusion in the abstract is valid from the main text. However, it is not obvious from the abstract. Some statement on differences between techniques should be added to the results section of the abstract.
Material and method
- l107: I guess it should be "appropriate"
- Table 1: Please carefully check all citations. For example, Vicini^78 but in the reference list 78=Haviland.
- Table 1: strange abbreviation "y trial"
- Table 1: What does T<1-1.4>cm mean?
- In general, the material & methods section could benefit from more details: How were hand searches performed (l81)? How were hazard ratios estimated (l91)? Which results depend on hazard ratio estimations? Mentioning that arms in Vicini were split for Fig. 2. However, I understand this has to be balanced with conciseness.
Results
- l155: it is unclear why the pooling IBTR+LRFS has been done. Was the reason to take the primary endpoints of each study? Why have not all studies been included in Appendix Fig 2?
- Is there still heterogeneity in IBTR after excluding IORT trials?
- Is the OR still significant after excluding IORT trials?
- Is the OR from IORT trials significantly different to the OR from other trials?
- Appendix Fig. 3: Colours for Orecchia vs Vaidya were difficult to distinguish, and even worse for Coles vs Meattini
- l185: "shows"
- Fig 2: As I understand the methods, Vicini was included twice without correction for the fact that both arms compare to the same comparison group (WBI). I don't expect this to flaw any result but the Total OR of Fig 2 then is technically not correct. This should be noted or the Total OR in Fig 2 removed (it repeats anyway Fig 1).
- Fig 6: I found this figure to be shifted to the appendix.
- l223: DMFI 2.8% vs 2.6%. Isn't the more common definition 97.2% vs 97.4% analogous to DFS below? This ambiguity makes some statements difficult to understand (direction of "increase"?). This is a general comment not limited to this paragraph.
- l233: "based"
- Table 4: Different trials are used for the different subgroups. However, there is heterogeneity between trials (Fig 1). Therefore, differences between subgroups may originate in the inclusion of different trials. For example the subgroup "Age>70" includes Orecchia which is likely to show higher OR due to another technique. The p-values for interaction are therefore to be interpreted as an exploratory analysis. Of course, this issue does not refer to the ASTRO risk groups comparison.
Discussion
- l348: "considered"
- l429: "this trial" ->meta analysis
Author Response
Reviewer 2
Dear authors,
overall, your manuscript is well written, the analysis is sound, conclusions are adequate, and the topic is interesting. Nevertheless, there are several points that should be improved.
Abstract
- "IBTR was significantly different between.." Please state the direction of the difference.
We have changed the line stating the direction of the difference.
- The conclusion in the abstract is valid from the main text. However, it is not obvious from the abstract. Some statement on differences between techniques should be added to the results section of the abstract.
We have added some details regarding the differences between the techniques.
Material and method
- l107: I guess it should be "appropriate"
We have changed the line accordingly.
- Table 1: Please carefully check all citations. For example, Vicini^78 but in the reference list 78=Haviland.
We have checked and updated all citations.
Table 1: strange abbreviation "y trial"
We have changed the abbreviation.
- Table 1: What does T<1-1.4>cm mean?
We have changed the abbreviation.
- In general, the material & methods section could benefit from more details: How were hand searches performed (l81)? How were hazard ratios estimated (l91)? Which results depend on hazard ratio estimations? Mentioning that arms in Vicini were split for Fig. 2. However, I understand this has to be balanced with conciseness.
We have updated the paragraph to be more precise in the mentioned aspects.
Results
- l155: it is unclear why the pooling IBTR+LRFS has been done. Was the reason to take the primary endpoints of each study? Why have not all studies been included in Appendix Fig 2?
Several studies only reported on LRFS using hazard ratios. Some studies used hazard ratios on IBTR. We wanted to analyze the primary endpoints of the trials in the most appropriate way (2 graphs: 1 for Odds ratio, 1 for Hazard ratio (IBTR and LRFS).
- Is there still heterogeneity in IBTR after excluding IORT trials?
No, the test for heterogeneity is no significant (p=0.625). We added this in the manuscript.
- Is the OR still significant after excluding IORT trials?
Yes, OR=1.367 CI95%: 1.103-1.695
- Is the OR from IORT trials significantly different to the OR from other trials?
Yes, the heterogeneity test for the effect size of PBI vs. WBI between BT+EBRT vs. WBI and IORT vs. WBI is significant.
- Appendix Fig. 3: Colours for Orecchia vs Vaidya were difficult to distinguish, and even worse for Coles vs Meattini
We have changed the colors to allow more distinction.
- l185: "shows"
We have changed the line accordingly.
- Fig 2: As I understand the methods, Vicini was included twice without correction for the fact that both arms compare to the same comparison group (WBI). I don't expect this to flaw any result but the Total OR of Fig 2 then is technically not correct. This should be noted or the Total OR in Fig 2 removed (it repeats anyway Fig 1).
We have mentioned this in the graph description.
- Fig 6: I found this figure to be shifted to the appendix.
We have added Figure 6 to the manuscript.
- l223: DMFI 2.8% vs 2.6%. Isn't the more common definition 97.2% vs 97.4% analogous to DFS below? This ambiguity makes some statements difficult to understand (direction of "increase"?). This is a general comment not limited to this paragraph.
We have changed this throughout the manuscript
- l233: "based"
Change done.
- Table 4: Different trials are used for the different subgroups. However, there is heterogeneity between trials (Fig 1). Therefore, differences between subgroups may originate in the inclusion of different trials. For example the subgroup "Age>70" includes Orecchia which is likely to show higher OR due to another technique. The p-values for interaction are therefore to be interpreted as an exploratory analysis. Of course, this issue does not refer to the ASTRO risk groups comparison.
We have mentioned this in the discussion, but we also think that it should be more highlighted. We have added some details in the results section.
Discussion
- l348: "considered"
We have changed this.
- l429: "this trial" ->meta analysis
We have changed this.
